# Blood co-expression modules identify potential modifier genes of diabetes and lung function in cystic fibrosis

**Fanny Pineau**[1], **Davide Caimmi**[2], **Milena Magalhães**[1], **Enora Fremy**[1],
**Abdillah Mohamed**[1], **Laurent Mely**[3], **Sylvie Leroy**[4], **Marlène Murris**[5], **Mireille Claustres**[1,6],
**Raphael Chiron**[2], **Albertina De Sario**[1] *

**1** EA7402, Laboratoire de Génétique de Maladies Rares (LGMR), University of Montpellier, Montpellier, France, **2** CRCM, Arnaud de Villeneuve Hospital, Montpellier, France, **3** CRCM, Renée Sabran Hospital, Hyères, France, **4** CRCM, Pasteur Hospital, Nice, France, **5** CRCM, Larrey Hospital, Toulouse, France, **6** CHU Montpellier, Laboratoire de Génétique Moléculaire, Montpellier, France

* albertina.de-sario@inserm.fr

**Data Availability Statement:** Normalized data and raw data generated during the current study are available in Gene Expression Omnibus (GEO) with

## Abstract

Cystic fibrosis (CF) is a rare genetic disease that affects the respiratory and digestive systems. Lung disease is variable among CF patients and associated with the development of comorbidities and chronic infections. The rate of lung function deterioration depends not only on the type of mutations in *CFTR*, the disease-causing gene, but also on modifier genes. In the present study, we aimed to identify genes and pathways that (i) contribute to the pathogenesis of cystic fibrosis and (ii) modulate the associated comorbidities. We profiled blood samples in CF patients and healthy controls and analyzed RNA-seq data with Weighted Gene Correlation Network Analysis (WGCNA). Interestingly, lung function, body mass index, the presence of diabetes, and chronic *P. aeruginosa* infections correlated with four modules of co-expressed genes. Detailed inspection of networks and hub genes pointed to cell adhesion, leukocyte trafficking and production of reactive oxygen species as central mechanisms in lung function decline and cystic fibrosis-related diabetes. Of note, we showed that blood is an informative surrogate tissue to study the contribution of inflammation to lung disease and diabetes in CF patients. Finally, we provided evidence that WGCNA is useful to analyze–omic datasets in rare genetic diseases as patient cohorts are inevitably small.

## Introduction

Cystic fibrosis (CF; OMIM 219700) is an autosomal recessive inherited disease that affects approximately 1/3000 newborns [1]. It results from impairment of the Cystic Fibrosis Transmembrane Conductance Regulator (CFTR) protein, a chloride channel expressed at the apical membrane of various epithelial cells. The defective protein results in thick, sticky and obstructive mucus in multiple organs of the respiratory, digestive and reproductive systems [1,2]. The mutant CFTR protein is also responsible for an altered innate and adaptive immune function.

accession number 136371 (https://www.ncbi.nlm.nih.gov/geo/query/acc.cgi?acc=GSE136371).

**Funding:** This study was supported by Vaincre La Mucoviscidose (RC20130500857, RC20140501072, RC20150501383, RC20170501949), Fondation Maladies Rares, and Montpellier Hospital to AD. FP was supported by Montpellier University and Montpellier Hospital, MMa by the Ciência Sem Fronteiras program (CNPq Brazil), and EF by Vaincre La Mucoviscidose. The funders had no role in study design, data collection and analysis, decision to publish, or preparation of the manuscript.

**Competing interests:** The authors have declared that no competing interests exist.

Lung disease is the main cause of morbidity and mortality in cystic fibrosis [2]. CF patients present chronic infections and abnormal inflammation of the lungs that lead to progressive airway destruction. The rate of lung function deterioration is variable among patients and associated with the development of comorbidities and chronic infections [1,2].

CF-related diabetes (CFRD) is a common comorbidity of CF [3]. It affects about 20% of adolescents and 40% to 50% of adults, and is associated with more frequent pulmonary exacerbations, accelerated pulmonary function decline and higher mortality. CFRD is characterized by a reduced and delayed insulin response. The beta-cell dysfunction is evident before the onset of diabetes and is already associated with a pulmonary function decline [3]. The exact causes of CFRD are not totally elucidated, nor is explained the association between diabetes and accelerated lung function loss. Mutant cftr-/- newborn zebrafishes have fewer beta cells than the wildtype ones, which suggests that the CFTR protein is important for pancreas development [4]. In addition, CFTR seems to be critical for insulin exocytosis, which implies that CF patients have an intrinsic pancreatic islet dysfunction [5]. Finally, the continuous infiltration of immune cells into the pancreas may contribute to the progressive destruction of the islets [6].

The defective CFTR protein and subsequent insufficient mucociliary clearance predispose CF patients to acute and, ultimately, chronic lung infections with opportunistic pathogens [7]. Chronic *P. aeruginosa* infection is found in approximately 40% of adult CF patients and is also associated with a drastic decrease of lung function [7].

Previous genetic studies showed that the clinical variability of CF patients depends not only on the type of mutations in the *CFTR* gene, but also on modifier genes, other genes that modulate the patient phenotype [1]. Much current research focuses on finding CF modifier genes to develop new therapies [1].

In the present study, we analyzed the transcriptome to identify genes and pathways that (i) contribute to the pathogenesis of cystic fibrosis and (ii) modulate the associated comorbidities. Knockout mice models have limitations because they do not develop spontaneous diabetes, nor do they present lung disease [8]; on the other hand, human pancreas and airway tissues are not easily accessible for studies. Herein, we used whole blood samples from CF patients as a surrogate tissue. Of interest, we found that lung function, body mass index (BMI), the presence of diabetes, and chronic *P. aeruginosa* infection correlated with modules of co-expressed genes.

## Materials and methods

### Study population

The study population included $\geq$ 18-year old subjects from the MethylCF cohort: 33 cystic fibrosis patients and 16 healthy controls [9,10]. A replication set of subjects from the same cohort was used for real-time PCR validation (20 CF patients and 8 healthy controls). The two sets were similar with respect to the age and male-to-female ratio. Demographic and clinical features are reported (**Table 1**). CF patients carried the homozygous p.Phe508del mutation. The presence of diabetes was determined on the basis of an abnormal oral glucose-tolerance test. CF patients were classified as chronically infected by P. *aeruginosa*, methicillin-resistant S. *aureus* and/or A. *fumigatus*, whenever they had three consecutive positive sputum cultures after antibiotic treatment. The study was approved by the "Comité de Protection des Personnes Sud Méditerranée III" Institutional Review Board (2013.02.01bis) and is registered at clinical.gov under reference #NCT02884. Informed written consent was obtained from all participants.

**Table 1. Demographic and relevant clinical features of the MethylCF cohort.**

| | Discovery set | | Replication set | |
|---|---|---|---|---|
| | **Controls** | **CF patients** | **Controls** | **CF patients** |
| | **(n = 16)** | **(n = 33)** | **(n = 8)** | **(n = 20)** |
| Age, years† | 28 | 28 (10) | 31 | 25 (8) |
| Sex, M:F | 10:6 | 23:10 | 3:5 | 10:10 |
| BMI, kg/m$^2$† | | 21 (4) | | 21 (4) |
| Weight, kg† | | 60 (13) | | 60 (12) |
| Height, cm† | | 171 (9) | | 168 (11) |
| FEV$_1$, %† | | 48 (24) | | 48 (24) |
| FEV$_1$, L† | | 1.91 (1.1) | | 1.70 (0.8) |
| FVC, %† | | 76 (22) | | 68 (28) |
| FVC, L† | | 3.32 (1.1) | | 2.99 (0.6) |
| PI, % | | 100 | | 100 |
| Diabetes, %* | | 36 | | 30 |
| HbA1c, %† | | 6.1 (1.1) | | 5.4 (na) |
| Atopy, % | | 18 | | 45 |
| *P. aeruginosa*, %§ | | 94 | | 95 |
| *MRSA*, %§ | | 36 | | 15 |
| *A. fumigatus*, % | | 21 | | 25 |
| Azythromycin, % | | 91 | | 100 |
| Aztreonam, % | | 12 | | 25 |
| Colistin, % | | 39 | | 50 |
| Tobramycin, % | | 55 | | 50 |
| Corticosteroid, % | | 33 | | 45 |

BMI, body mass index; CF, cystic fibrosis; FEV$_1$, forced expiratory volume in 1 second

FVC, forced vital capacity; HbA1c, glycated hemoglobin fraction; PI, pancreatic insufficiency

MRSA, Methicillin-resistant *Staphyloccocus aureus*.

† Median values (interquartile range).

* The presence of diabetes was determined on the basis of an abnormal oral glucose-tolerance test.

§ CF patients were classified as chronically infected by P. *aeruginosa*, MSRA and/or A. *fumigatus*

whenever they had three consecutive positive sputum cultures after antibiotic treatment.

na, not applicable because only five measurements were available.

## RNA sequencing and differential expression analysis

RNA was extracted from whole blood samples using the PAXgene Blood RNA kit (#762124, PreAnalytix), according to the manufacturer's recommendations [9]. Total RNA sequencing libraries were prepared with the TruSeq Stranded Total RNA kit (Illumina®) and ribosomal RNA was depleted using the Ribo-Zero Gold rRNA removal kit following the manufacturer's instructions. Libraries were sequenced in paired-end 75 nucleotides mode with a HiSeq4000 Illumina. The quality of raw sequenced reads was assessed using the FASTQC quality control tool and reads were mapped to the reference human genome build hg19/GRCh37 with Tophat 2 [11]. We used HTSeq to obtain the number of reads associated with each gene in the Gencode v26lift37 database (restricted to protein-coding genes, antisense and lincRNAs) [12]. The differential expression of the annotated genes was calculated using DESeq [13]. Transcripts with a minimum 2-fold change and a Benjamin-Hochberg adjusted p-value (FDR) < 0.05 were considered as differentially expressed. Normalized data and raw data generated during

the current study are available in Gene Expression Omnibus (GEO) with accession number 136371 (https://www.ncbi.nlm.nih.gov/geo/query/acc.cgi?acc=GSE136371).

## Real-time PCR validation

Total RNA from whole blood samples were reverse transcribed from 500 ng as previously described [9]. Primers were designed using Primer3Plus and Beacon Designer Free Edition online tools, using a Gibbs energy threshold of $\Delta G \geq -2.0$ for hetero- and homodimers and a GC content $> 40\%$ (S1 Table). All primers were tested to display an efficiency of amplification of at least 93% (±SD 6%). Amplicons overlapped exon junctions except for CITF22-49E9.3. Real-time PCR reactions were done in duplicate in two independent reverse transcriptions, using SYBR Green I Master mix (Roche Diagnostics) and a LightCycler 480 Instrument. The reverse transcription reaction program consisted of 10 min pre-incubation at 95˚C followed by a three step amplification (95˚C for 10 s, 60˚C for 20 s, 72˚C for 10 s). Standard curves were generated by serial dilutions of a control cDNA. Expression levels were expressed as ratios relative to that of the reference gene (*YWHAZ*), using the Pffafl method (with efficiency correction) [14]. Differences between groups were analyzed with Wilcoxon test and were considered significant when p-value $< 0.05$.

## Weighted Gene Correlation Network Analysis

To identify modules of co-expressed genes, we implemented Weighted Gene Correlation Network Analysis in the WGCNA R package [15]. We used the WGCNA functions to (i) construct a network of coexpressed and highly connected genes, (ii) identify modules of coexpressed genes, and (iii) correlate the gene modules with biological features (continuous or binary phenotypic traits) of the MethylCF cohort. Unless otherwise specified below, we used the default parameters as described by [15].

Briefly, we preselected a list of 15077 genes with a FPKM $> 0.1$ and log2-transformed the FPKM values (FPKM value +1). Then, we generated a signed adjacency matrix using the biweight midcorrelation and raising it to the power beta = 18 to reduce the noise. The adjacency network exhibited approximate scale-free topology ($R^2 = 0.92$). Scale-free topology is obtained when few genes are highly connected to each other (hub genes), whereas the remaining genes are weakly connected. The adjacency matrix was transformed into a topological overlap matrix, an adjacency matrix that considers coexpression information and topological similarity. Modules were generated using the dynamic tree cut and modules with highly correlated module eigengenes (correlation $> 0.75$) were merged together. Correlations between the modules of coexpressed genes (eigengenes) and clinical and demographic features of the MethylCF cohort were calculated. Top genes in the modules were visualized with Cytoscape [16].

## Gene ontology analysis

Gene ontology (GO) and KEGG pathways were analyzed with WebGestalt (WEB-based GEne SeT AnaLysis Toolkit; URL: http://www.webgestalt.org) using the Benjamini–Hochberg correction for multiple testing [17]. For GO, we retained a false discovery rate of 5%, excluding categories with less than four genes. For KEGG pathway analyses, we used a false discovery rate of either 1% or 5%.

# Results

## RNA sequencing and biotype distribution

Whole blood samples had been collected from CF patients with no ongoing pulmonary exacerbation [9]. Total blood cell count and the percentage of different types of leukocytes were

within normal range. The median percentages calculated on 19 CF patients were 60% (iqr 14.7%) neutrophils, 26% (iqr 12.0%) lymphocytes, 9% (iqr 3.0%) monocytes, 3% (iqr 2.6%) eosinophils, 1% (iqr 0.5%) basophils. We collected the blood samples in PAXgene tubes that stabilize RNA, preserve all types of circulating cells (leukocytes and platelets) and do not alter the gene expression profile of frozen samples [18].

Using RNA-seq, we generated the transcriptome of 49 blood samples from the MethylCF cohort [9,10]. Eight samples were excluded because ribosomal depletion failed. Hence, the bioinformatic analyses were carried out on 41 samples (27 CF patients and 14 controls). The median total number of reads per sample was 112 million (iqr 17 million). We found that 9324 protein coding genes, 501 antisense transcripts and 493 lincRNA were expressed in blood samples (FPKM >1). Biotype distribution and expression level of the corresponding transcripts are represented in **S1 Fig**.

## Differentially expressed genes between CF patients and controls

When we compared gene expression between CF patients and controls, we found 75 differentially expressed (DE) genes (48 genes were over-expressed and 27 genes were under-expressed in CF patients) (Log$_2$foldChange $\geq$ 1 or $\leq$ -1; FDR < 0.05) (**Fig 1**, **S2 Table**). Thirteen non-coding RNAs were over-expressed and two non-coding RNAs were under-expressed in CF patients compared to controls (**S2 Table**).

Gene ontology (GO) analysis showed that DE genes between CF patients and controls were overrepresented among genes important for the response to bacterial infection (FDR p-value = 1.2E-05, 19 genes including *TLR5*, *S100A8*, *S100A12*, *ILR23R)* and leukocyte activation (FDR p-value = 3.4E-04, 11 genes including *IL23R*, *IL4R*, *CDC80*, *TBX21*) (**S3 Table**). KEGG analysis highlighted the Th17 lymphocyte activation pathway (FDR p-value = 4.6E-03, 7 genes: *MAPK14*, *IL23R*, *TBX21*, *HLA-DOA*, *IL2RB*, *IL4R*, *RORC)*.

## Validation of RNA-seq data with real-time PCR

To validate the RNA-seq data, we assessed the expression levels of three DE genes (*TLR5*, *CLEC4D*, and *ALPL*) and one DE lincRNA (CITF22-49E9.3) using real-time PCR in the same set of blood samples (n = 49) as a technical validation, and in a replication set (n = 28) as a biological validation (**Table 2**). We selected protein-coding and non-coding transcripts among the top DE genes with a relevant biological function and a range of expression levels (from 5 to 56 FPKM). *TLR5*, *CLEC4D*, *ALPL* and CITF22-49E9.3 were differentially expressed between CF patients and controls of the discovery set, and thus technically validated (p-value < 0.01). *TLR5*, *CLEC4D*, and *ALPL* were biologically validated since their expression levels differed between CF patients and controls of the replicative set (p-value < 0.05). CITF22-49E9.3 failed to be biologically validated (p-value = 0.18), but the direction was the same as in RNA-seq (over-expression in CF). For all loci, the direction of differential expression was identical and fold-changes were similar between RNA-seq and real-time PCR data, showing a total concordance between the two techniques (**Table 2**).

## Weighted Gene Correlation Network Analysis

Next, to find additional genes important for CF pathogenesis and the associated comorbidities, we implemented Weighted Gene Correlation Network Analysis (WGCNA) [15]. Using RNA-seq datasets from 27 CF patients of the MethylCF cohort, we found 28 modules of co-expressed genes (**S4 Table**). The number of genes in each module ranged from 35 to 2839. A majority of modules were enriched with genes that belong to biological pathways (Notch signaling, MAPK signaling, platelet activation, B cell receptor, etc), which suggests that the gene modules are biologically meaningful (**Table 3**).

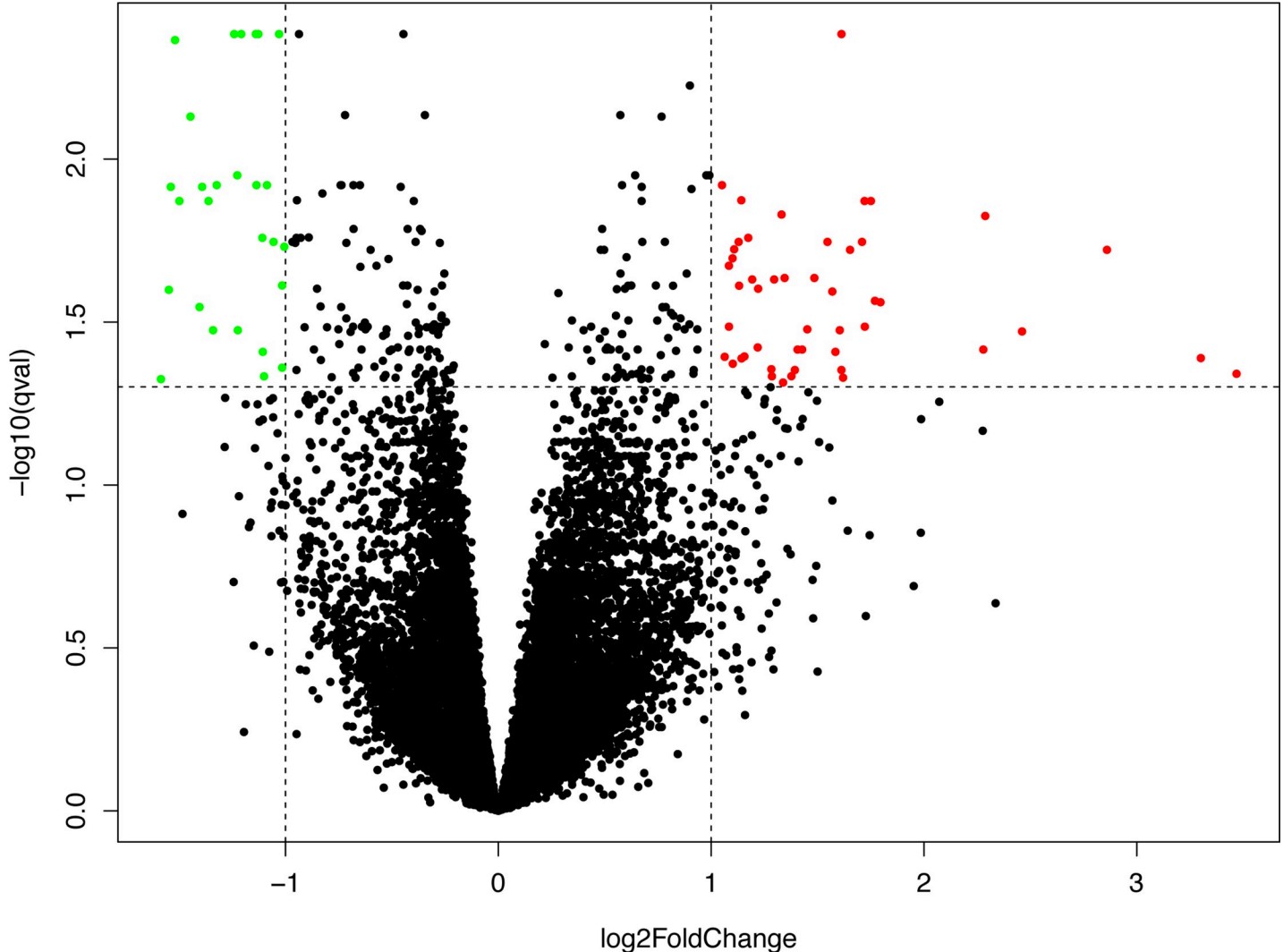

**Fig 1. Differentially expressed gene between CF patients and controls.** The volcano plot represents the $Log_2$ transformed fold-changes (x-axis) and the $Log_{10}$ transformed q-values (y-axis). 48 genes were over-expressed (red) and 27 genes were under-expressed (green) in CF blood samples (FDR < 0.05).

**Table 2. Validation of RNA-seq data by real time-PCR.**

| | RNA-seq—Discovery set | | | | real time-PCR—Discovery set | | | | real time-PCR—Replication set | | | |
|---|---|---|---|---|---|---|---|---|---|---|---|---|
| | FPKM | | FC | p-value† | Normalized ratios | | FC | p-value† | Normalized ratios | | FC | p-value† |
| | Med CF | Med C | | | Med CF | Med C | | | Med CF | Med C | | |
| **TLR5** | 13.87 | 4.69 | 2.96 | 3.9E-05 | 2.59 | 1.15 | 2.26 | 6.6E-05 | 2.72 | 1.39 | 1.96 | 3.9E-03 |
| **CLEC4D** | 14.70 | 8.73 | 1.68 | 3.5E-05 | 5.72 | 2.60 | 2.20 | 2.6E-06 | 4.68 | 2.61 | 1.79 | 1.6E-02 |
| **ALPL** | 49.21 | 19.32 | 2.55 | 7.9E-04 | 2.34 | 0.84 | 2.80 | 7.3E-03 | 2.88 | 0.91 | 3.18 | 2.1E-03 |
| **CITF22-49E9.3** | 9.35 | 5.38 | 1.74 | 6.1E-05 | 3.37 | 2.09 | 1.61 | 1.5E-06 | 3.15 | 2.20 | 1.43 | 1.8E-01 |

FPKM: Fragments Per Kilobase of transcript per Million mapped reads, Med: median, CF: Cystic Fibrosis, C: Control, FC: fold-change (CF/C).

† CF vs C, Wilcoxon test.

Next, we calculated the correlation between the clinical traits of the patients and the eigengenes of the modules (**Fig 2**). The eigengene is the first component of a principal component analysis and represents the summary of the gene expression profile of the module [15]. Modules of co-expressed genes that correlated with lung function, the presence of diabetes, and a chronic *P. aeruginosa* infection were analyzed in more detail.

Lung function metrics included forced expiratory volume in 1 second ($FEV_1$) and forced vital capacity (FVC) expressed in liters and as a percent predicted based on age, sex and height [19]. These clinical measures are used for the follow-up of CF patients and as endpoints to assess whether patients respond to treatments [20]. In the MethylCF cohort, lung function and BMI best correlated with the dark turquoise module. For simplicity, only $FEV_1$ (%) was represented (**Fig 2**), however, consistent results were obtained with FVC (%) (r = 0.54 p-value = 4.0E-03). The GO analysis of the dark turquoise module highlighted terms related to cell-cell adhesion and cell adherence junctions (**Table 4**). *FLNA* is a hub gene of this module. It encodes Filamin A, an actin-binding and scaffolding protein that interacts with integrins to regulate leukocyte trafficking (**Fig 3**) [21].

Diabetes and glycated hemoglobin (HbA1c) levels correlated with the light yellow module. HbA1c reflects the mean glucose levels over a three-month period. GO analysis of this module revealed terms related to vesicle transport and platelet activation (**Table 4**). Among the hub genes of this network were integrin genes (*ITGB3*, *ITGA2B*, *ITGB5*) (**Fig 3**).

The presence of diabetes (but not HbA1c levels) also correlated with the cyan module. GO analysis of the genes belonging to this module revealed enrichment for terms related to the proteasome (**Table 5**). Through hierarchical clustering, we identified three groups of CF patients with distinct gene expression signatures and prevalence of diabetes (**Fig 4**). In the group enriched with diabetic patients (6 out of 9 patients), seven proteasome genes (*PSME4*, *PSMA7*, *PSMB4*, *PSMB6*, *PSMC4*, *PSMD7* and *PSMD8*) and three genes previously associated with common diabetes (*SNX17*, *PARK7/DJ-1* and *ATP5B)* were highly expressed (**Fig 4**). By contrast, two genes encoding histone methyltransferases (*SMYD3* and *KMT2A*) and one gene encoding a chromatin-remodeling ATPase (*EP400)* were under-expressed in this group of patients.

Finally, the presence of a chronic *P. aeruginosa* infection negatively correlated with the eigengene of the magenta module. GO analysis revealed terms related to heme metabolic process and hemoglobin complex (**Table 5**). The magenta network comprised three hub genes encoding the following proteins: SLC25A39 (Solute Carrier Family 25 member 39), a mitochondrial solute carrier protein, GATA1 (GATA Binding Protein 1), an erythroid transcription factor and CDC34 (Cell Division Cycle 34), a ubiquitin-conjugating enzyme (**S2 Fig**).

## Discussion

CF patients present a clinical heterogeneity that is not fully explained by the type of mutations in the *CFTR* gene. Previous studies emphasized that other genes modulate the clinical phenotype and account for the development of comorbidities [1]. CF modifier genes were extensively searched in genetic studies first and more recently in transcriptomic studies [1].

Transcriptomic analyses in airway cell lines and nasal epithelial cell samples showed expression changes in genes involved in cell proliferation, inflammation and immune responses, protein metabolism, and calcium and membrane pathways [22–24]. More recently, blood samples from CF patients presenting either severe or mild lung disease were analyzed: genes of the type I interferon response and ribosomal stalk proteins were differentially expressed [25]. However, no healthy subjects were compared with CF patients in that study.

Herein, we used RNA-seq to profile blood samples from CF patients and healthy controls. An added value of this cohort is that the clinical data were recorded on the same day biological

**Table 3. Module of coexpressed genes: KEGG analysis.**

| | Size* | KEGG § | Pathway | N | R | FDR |
|---|---|---|---|---|---|---|
| Darkgreen | 90 | | | | | n.s. |
| Red | 802 | hsa04330 | Notch signaling pathway | 48 | 6.9 | 4.3E-04 |
| Turquoise | 2838 | hsa04666 | Fc gamma R-mediated phagocytosis | 91 | 6.7 | 8.0E-10 |
| Black | 679 | hsa00310 | Lysine degradation | 59 | 6.5 | 4.2E-03 |
| Tan | 243 | hsa05210 | Colorectal cancer | 86 | 7.8 | 1.8E-02 |
| White | 35 | hsa04120 | Ubiquitin mediated proteolysis | 137 | 10.2 | 6.8E-03 |
| Orange | 66 | hsa04217 | Necroptosis | 162 | 12.1 | 1.3E-06 |
| Royalblue | 140 | hsa04621 | NOD-like receptor signaling | 168 | 8.6 | 1.8E-06 |
| Darkred | 107 | | | | | n.s. |
| Greenyellow | 320 | hsa04932 | Non-alcoholic fatty liver disease | 149 | 7.1 | 2.5E-06 |
| Purple | 573 | hsa04144 | Endocytosis | 244 | 3.3 | 3.2E-04 |
| LightYellow | 143 | hsa04611 | Platelet activation | 122 | 8.2 | 9.3E-05 |
| Magenta | 595 | | | | | n.s. |
| Salmon | 235 | | | | | n.s. |
| Cyan | 173 | hsa03050 | Proteasome | 44 | 20.4 | 1.0E-05 |
| Darkgrey | 68 | hsa05203 | Viral carcinogenesis | 201 | 14.2 | 0.0E+00 |
| Lightgreen | 145 | | | | | n.s. |
| Pink | 673 | hsa04010 | MAPK signaling | 255 | 2.8 | 1.0E-03 |
| Darkturquoise | 79 | | | | | n.s. |
| Grey60 | 156 | hsa01230 | Biosynthesis of amino acids | 75 | 8.2 | 6.7E-03 |
| Darkorange | 59 | | | | | n.s. |
| Midnightblue | 171 | hsa04650 | Natural killer cell mediated cytotoxicity | 131 | 6.3 | 0.0E+00 |
| Brown | 1484 | hsa03010 | Ribosome | 154 | 12.2 | 0.0E+00 |
| Green | 1097 | hsa03010 | Ribosome | 153 | 6.4 | 1.6E-12 |
| Lightcyan | 169 | hsa04662 | B cell receptor signaling | 71 | 16.2 | 7.8E-06 |
| Blue | 1859 | hsa04660 | T cell receptor signaling | 101 | 4.9 | 2.6E-02 |
| Yellow | 1320 | | | | | n.s. |
| Grey | 757 | | | | | n.s. |

*Number of genes in the module

§ Top KEGG pathway

N, number of genes in the pathway. R, ratio of enrichment. FDR, false discovery rate

n.s., not significant, p-value > 0.05 after Benjamini-Hochberg correction

samples were collected. Importantly, in addition to the differential gene expression analysis, we implemented WGCNA.

DE genes between CF patients and controls were overrepresented among genes important for the leukocyte activation and the response to bacterial infection, which is consistent with the permanent inflammation and chronic infections in CF patients. KEGG analysis highlighted the Th17 lymphocytes activation pathway.

A limitation of the DE gene analysis is that a number of relevant genes do not reach significance after correction for multiple tests, unless large cohorts are assembled. But it is difficult to fulfill this condition in rare disease studies. To overcome this drawback, we set up WGCNA [15]. WGCNA detects networks of co-regulated and highly connected genes that belong to biological pathways and reduces the number of variables to be tested, thus decreasing the false discovery rate [15]. Using WGCNA, we found that clinical traits of interest in cystic fibrosis correlated with modules of co-expressed genes in blood samples.

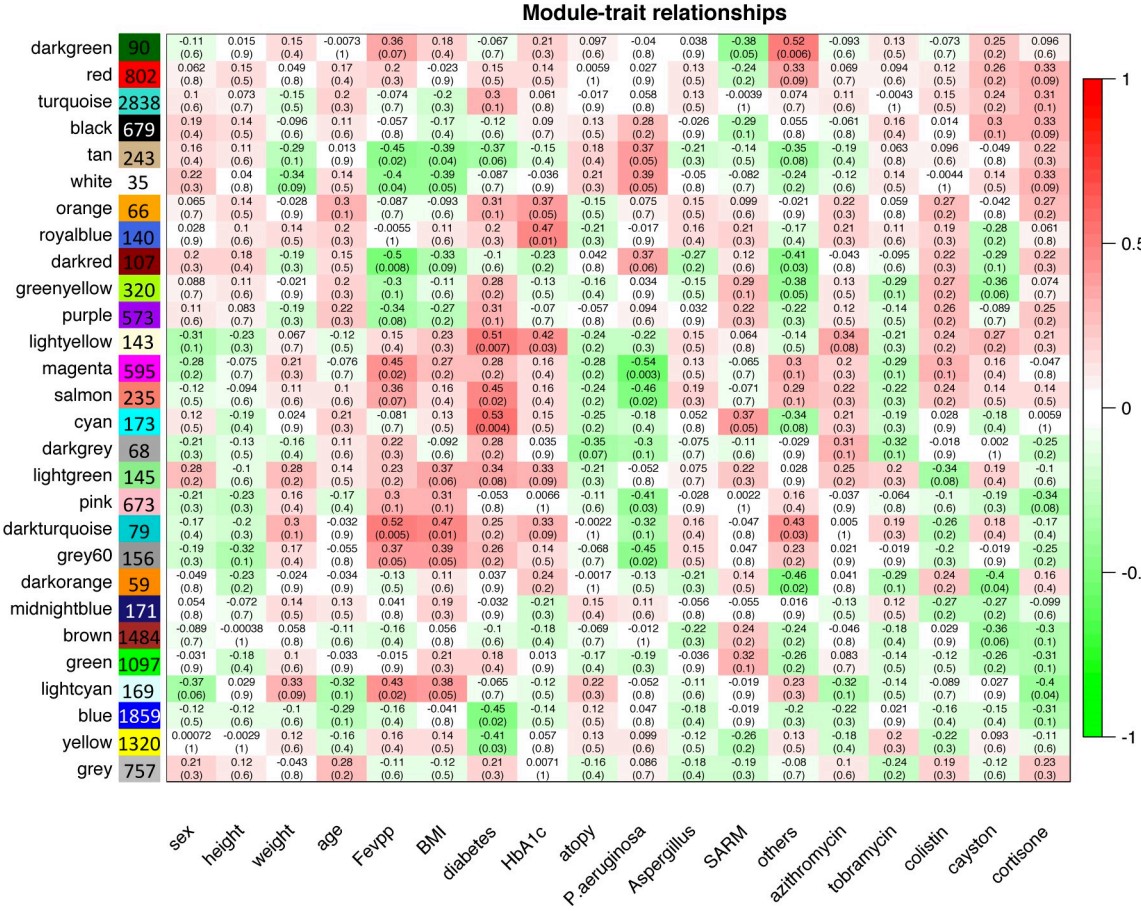

**Fig 2. WGCNA on blood RNA-seq dataset.** The heatmap represents the correlation (coefficient and p-value) between eigengene modules and clinical traits. Module sizes are shown in the colored boxes.

Because lung disease is the main cause of mortality and morbidity in CF, first we inspected the dark turquoise module that correlated with the lung function ($FEV_1$ and FVC) and BMI. Lung function and BMI are positively correlated in CF and their deterioration is predictive of patient decline and, ultimately, patient death [26]. The GO analysis of the dark turquoise module showed terms related to cell-cell adhesion and cell adherence junctions. Interestingly, the same association between lung function and cell adhesion was highlighted by the DNA methylation analysis of CF nasal epithelial cell samples [10].

In the present transcriptomic study, *FLNA* is a hub gene of the dark turquoise module. The Filamin A protein is an actin-binding and scaffolding protein that binds to integrins and also interacts with CFTR [27]. Of interest, mutations in the *FLNA* gene result in interstitial lung disease, a severe respiratory illness [28]. The FLNA protein is required for optimal T cell homing into lymph nodes and inflamed tissues [29]. To explain the association between lung function and cell adhesion in cystic fibrosis, we argue that if cell junctions are loosened, the leukocyte trafficking from the blood stream to the airways is facilitated, and the resulting high inflammation reduces lung function.

The decline of lung function is steeper in CF patients with diabetes and the fast decay starts 1–3 years before the appearance of diabetes [3]. Peaks of hyperglycemia also occur before the appearance of diabetes [3]. To explain the association between diabetes and a more rapid lung

**Table 4. GO terms for dark turquoise module correlated with lung function and BMI, and light yellow module correlated with diabetes.**

| Geneset | Description | Number of genes | Ratio of enrichment | FDR |
|---|---|---|---|---|
| **DARKTURQUOISE MODULE** | | | | |
| *BIOLOGICAL PROCESS* | | | | n.s. |
| *CELLULAR COMPONENT** | | | | |
| GO:0030529 | intracellular ribonucleoprotein complex | 12 | 4.4 | 7.7E-03 |
| GO:1990904 | ribonucleoprotein complex | 12 | 4.4 | 7.7E-03 |
| GO:0005912 | adherens junction | 11 | 4.3 | 1.0E-02 |
| GO:0061695 | transferase complex, transferring phosphorus-containing groups | 7 | 7.5 | 1.0E-02 |
| GO:0070161 | anchoring junction | 11 | 4.2 | 1.0E-02 |
| GO:0005730 | nucleolus | 12 | 3.8 | 1.1E-02 |
| GO:1990234 | transferase complex | 11 | 4.0 | 1.2E-02 |
| GO:0044798 | nuclear transcription factor complex | 5 | 10.1 | 1.7E-02 |
| GO:0005913 | cell-cell adherens junction | 7 | 5.9 | 2.0E-02 |
| GO:1902494 | catalytic complex | 14 | 3.0 | 2.0E-02 |
| *MOLECULAR FUNCTION* | | | | |
| GO:0003723 | RNA binding | 17 | 2.8 | 3.8E-02 |
| GO:0044877 | macromolecular complex binding | 15 | 3.0 | 3.8E-02 |
| GO:0098641 | cadherin binding involved in cell-cell adhesion | 7 | 6.5 | 3.8E-02 |
| GO:0098632 | protein binding involved in cell-cell adhesion | 7 | 6.2 | 3.8E-02 |
| GO:0044822 | poly(A) RNA binding | 14 | 3.1 | 3.8E-02 |
| GO:0098631 | protein binding involved in cell adhesion | 7 | 6.1 | 3.8E-02 |
| GO:0045296 | cadherin binding | 7 | 6.1 | 3.8E-02 |
| *KEGG PATHWAY* | | | | n.s. |
| **LIGHT YELLOW MODULE** | | | | |
| *BIOLOGICAL PROCESS** | | | | |
| GO:0007596 | blood coagulation | 34 | 4.2 | 6.8E-09 |
| GO:0042060 | wound healing | 43 | 3.4 | 6.8E-09 |
| GO:0050817 | coagulation | 34 | 4.1 | 6.8E-09 |
| GO:0007599 | hemostasis | 34 | 4.1 | 6.8E-09 |
| GO:0009611 | response to wounding | 44 | 2.9 | 3.4E-07 |
| GO:0030168 | platelet activation | 20 | 5.1 | 3.5E-06 |
| GO:0050878 | regulation of body fluid levels | 35 | 2.9 | 1.4E-05 |
| GO:0002576 | platelet degranulation | 15 | 6.0 | 3.2E-05 |
| GO:0070527 | platelet aggregation | 10 | 7.2 | 9.3E-04 |
| GO:0006887 | exocytosis | 27 | 2.8 | 1.4E-03 |
| *CELLULAR COMPONENT** | | | | |
| GO:0031091 | platelet alpha granule | 16 | 11.4 | 4.9E-10 |
| GO:0031410 | cytoplasmic vesicle | 73 | 2.3 | 6.1E-09 |
| GO:0097708 | intracellular vesicle | 73 | 2.3 | 6.1E-09 |
| GO:0044433 | cytoplasmic vesicle part | 49 | 2.8 | 8.4E-09 |
| GO:0031093 | platelet alpha granule lumen | 12 | 11.7 | 6.4E-08 |
| GO:0099503 | secretory vesicle | 28 | 3.3 | 8.0E-06 |
| GO:0034774 | secretory granule lumen | 12 | 7.6 | 8.4E-06 |
| GO:0030141 | secretory granule | 23 | 3.5 | 3.0E-05 |
| GO:0005925 | focal adhesion | 24 | 3.3 | 3.9E-05 |
| GO:0005924 | cell-substrate adherens junction | 24 | 3.3 | 3.9E-05 |
| *MOLECULAR FUNCTION* | | | | n.s. |
| *KEGG PATHWAY* | | | | |
| hsa04611 | Platelet activation—Homo sapiens (human) | 10 | 8.2 | 9.3E-05 |

* Top 10 enriched terms

n.s., not significant after Benjamini-Hochberg correction

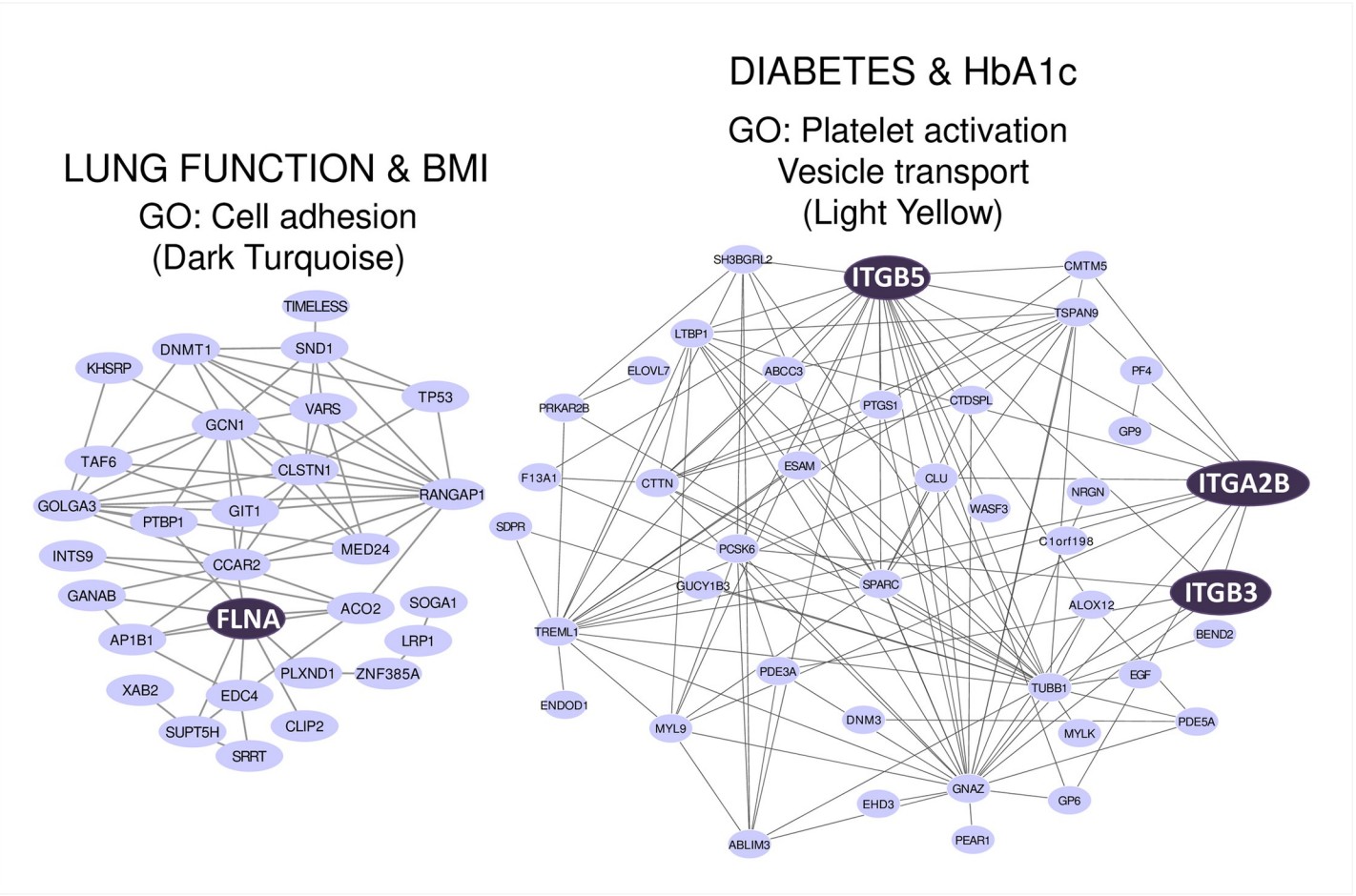

**Fig 3. Network representation of dark turquoise and light yellow modules.** Top genes and their connections were visualized with Cytoscape [16]. Genes of interest were emphasized. For each module the correlated clinical traits and main gene ontology (GO) terms are shown.

function decline, modules that correlated with lung function (dark turquoise) and with diabetes and HbA1c levels (light yellow) should be investigated together. Of interest, some of their respective hub genes encode proteins (FLNA and integrins) that bind one to each other to regulate T lymphocyte and neutrophil trafficking [26,30]. Also, high glucose modifies the levels of the Flna protein in rat endothelial cells [31]. All together, these findings suggest that glucose fluctuations can be the initial event that alter the expression of genes responsible for leukocyte trafficking, increases airway inflammation and, thereby, reduces the lung function in cystic fibrosis.

CF patients with diabetes may present microvascular complications, namely retinopathy and nephropathy [3]. Platelets are activated by glucose and their abnormalities are the initial event responsible for microvascular complications in diabetes [32]. Genes encoding platelet aggregation proteins were overrepresented in the light yellow module which, therefore, should be analyzed in detail with respect to these comorbidities.

The presence of diabetes but not HbA1c levels correlated with the cyan module. It comprised genes that encode proteins of the 20S and 19S proteasome subunits. A pivotal function of the proteasome is to degrade the oxidized and misfolded proteins that are generated by oxidative stress [33]. Through visualization of the most connected genes of this module, we

**Table 5. GO terms for cyan module correlated with diabetes and magenta module correlated with *P. aeruginosa* infection.**

| Geneset | Description | Number of genes | Ratio of enrichment | FDR |
|---|---|---|---|---|
| **CYAN MODULE** | | | | |
| ***BIOLOGICAL PROCESS*** | | | | |
| GO:0042180 | cellular ketone metabolic process | 9 | 8.6 | 7.0E-04 |
| GO:0009308 | amine metabolic process | 7 | 12.0 | 7.0E-04 |
| GO:0006520 | cellular amino acid metabolic process | 10 | 5.8 | 1.9E-03 |
| GO:0038061 | NIK/NF-kappaB signaling | 6 | 11.5 | 2.6E-03 |
| GO:0007164 | establishment of tissue polarity | 6 | 10.6 | 3.2E-03 |
| GO:0001738 | morphogenesis of a polarized epithelium | 6 | 9.6 | 4.7E-03 |
| GO:0010608 | posttranscriptional regulation of gene expression | 10 | 4.7 | 4.9E-03 |
| ***CELLULAR COMPONENT**** | | | | |
| GO:0000502 | proteasome complex | 6 | 25.5 | 6.3E-05 |
| GO:1905369 | endopeptidase complex | 6 | 25.5 | 6.3E-05 |
| GO:1902494 | catalytic complex | 18 | 4.0 | 8.0E-05 |
| GO:1905368 | peptidase complex | 6 | 18.9 | 1.9E-04 |
| GO:0005844 | polysome | 4 | 25.5 | 3.7E-03 |
| GO:0030529 | intracellular ribonucleoprotein complex | 11 | 4.2 | 7.0E-03 |
| GO:1990904 | ribonucleoprotein complex | 11 | 4.2 | 7.0E-03 |
| GO:0005839 | proteasome core complex | 3 | 40.1 | 7.0E-03 |
| GO:0005838 | proteasome regulatory particle | 3 | 38.3 | 7.2E-03 |
| GO:0022624 | proteasome accessory complex | 3 | 33.7 | 9.6E-03 |
| ***MOLECULAR FUNCTION*** | | | | |
| GO:0003723 | RNA binding | 22 | 3.7 | 2.8E-05 |
| GO:0044822 | poly(A) RNA binding | 19 | 4.3 | 2.8E-05 |
| GO:0004298 | threonine-type endopeptidase activity | 3 | 37.6 | 3.0E-02 |
| GO:0070003 | threonine-type peptidase activity | 3 | 37.6 | 3.0E-02 |
| GO:0035257 | nuclear hormone receptor binding | 5 | 10.2 | 4.7E-02 |
| ***KEGG PATHWAY*** | | | | |
| hsa03050 | Proteasome—Homo sapiens (human) | 6 | 29.6 | 1.1E-05 |
| **MAGENTA MODULE** | | | | |
| ***BIOLOGICAL PROCESS**** | | | | |
| GO:0006778 | porphyrin-containing compound metabolic process | 9 | 12.9 | 1.7E-04 |
| GO:0016567 | protein ubiquitination | 38 | 2.6 | 3.2E-04 |
| GO:0030163 | protein catabolic process | 38 | 2.5 | 5.2E-04 |
| GO:0006779 | porphyrin-containing compound biosynthetic process | 7 | 15.1 | 5.2E-04 |
| GO:0030218 | erythrocyte differentiation | 12 | 6.5 | 5.5E-04 |
| GO:0033014 | tetrapyrrole biosynthetic process | 7 | 13.4 | 7.3E-04 |
| GO:0046501 | protoporphyrinogen IX metabolic process | 5 | 25.9 | 7.3E-04 |
| GO:0015669 | gas transport | 6 | 17.3 | 7.3E-04 |
| GO:0034101 | erythrocyte homeostasis | 12 | 6.0 | 7.3E-04 |
| GO:0032446 | protein modification by small protein conjugation | 39 | 2.3 | 7.9E-04 |
| ***CELLULAR COMPONENT**** | | | | |
| GO:0005833 | hemoglobin complex | 6 | 38.3 | 3.5E-06 |
| GO:0030863 | cortical cytoskeleton | 11 | 9.2 | 1.0E-05 |
| GO:0014731 | spectrin-associated cytoskeleton | 5 | 43.9 | 1.0E-05 |
| GO:0005768 | endosome | 31 | 2.7 | 9.1E-05 |
| GO:0005773 | vacuole | 26 | 2.9 | 2.4E-04 |
| GO:0044448 | cell cortex part | 11 | 6.3 | 2.4E-04 |

(*Continued*)

**Table 5.** (Continued)

| Geneset | Description | Number of genes | Ratio of enrichment | FDR |
|---------|-------------|-----------------|---------------------|-----|
| GO:0005856 | cytoskeleton | 54 | 1.9 | 3.4E-04 |
| GO:0036019 | endolysosome | 5 | 20.6 | 3.9E-04 |
| GO:0031410 | cytoplasmic vesicle | 48 | 1.9 | 5.1E-04 |
| GO:0097708 | intracellular vesicle | 48 | 1.9 | 5.1E-04 |
| *MOLECULAR FUNCTION* | | | | |
| GO:0046983 | protein dimerization activity | 41 | 2.10 | 9.4E-03 |
| *KEGG PATHWAY* | | | | n.s. |

Top 10 enriched terms

n.s., not significant after Benjamini-Hochberg correction

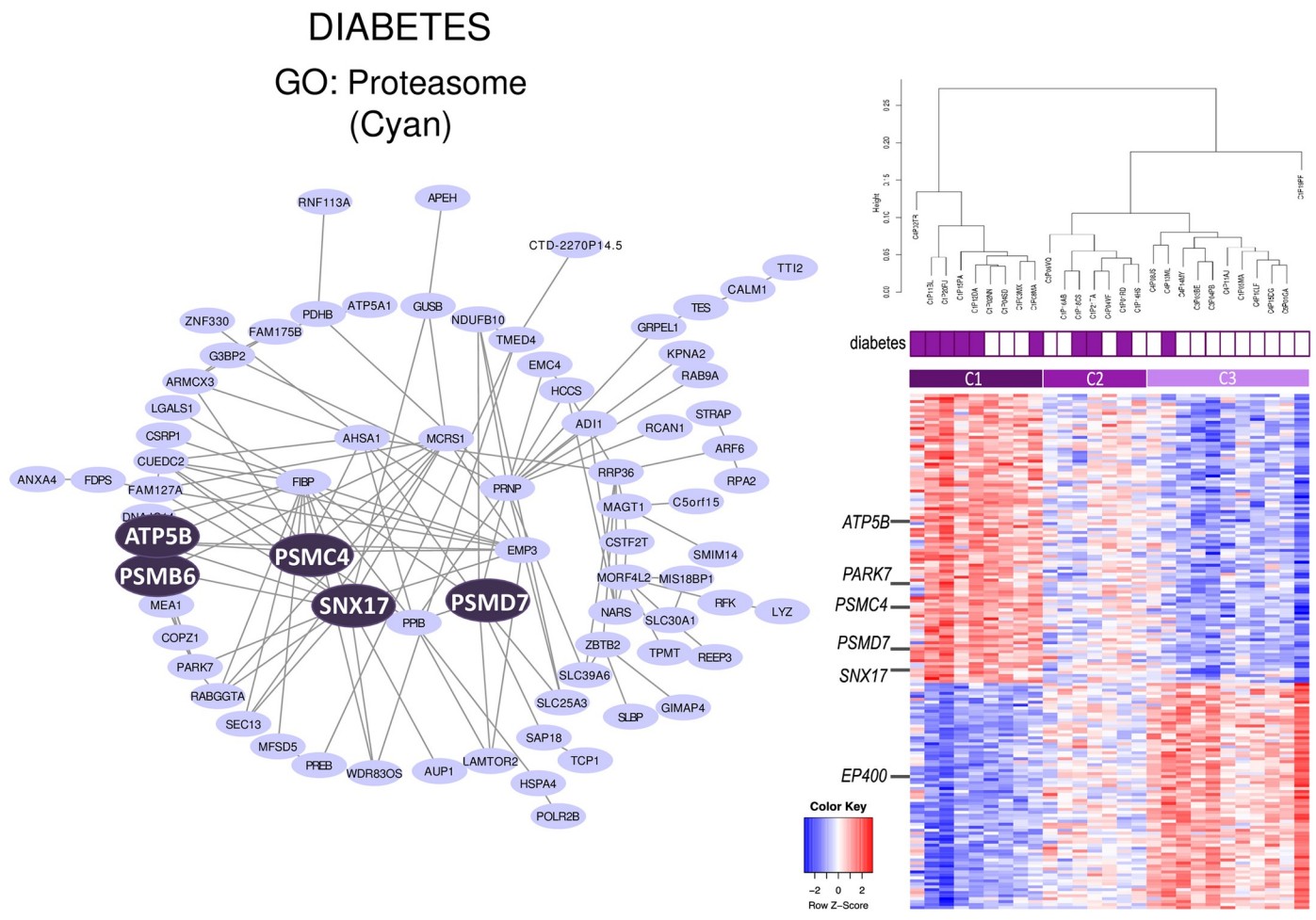

**Fig 4. Network representation of cyan module.** Top genes and their connections were visualized with Cytoscape [16] (left side). Genes of interest were emphasized. The correlated clinical trait and main gene ontology (GO) term are shown. Hierarchical clustering and heatmap of the cyan genes: purple square, CF patients with diabetes; white square, CF patients without diabetes. C1, C2, C3: cluster 1, 2 and 3, respectively (right side).

identified *SNX17*, previously associated with glucose-homeostasis in muscle and adipose tissues of type 2 diabetic patients [34]. The SNX17 protein activates T cells by regulating T cell receptors and integrin recycling in humans [35]. *SNX17* is a hub connected to 12 co-expressed genes, namely *PARK7/DJ-1* encoding a protein deglycase and *ATP5B* encoding the mitochondrial ATP synthase B subunit. In diabetic mice, the expression of the ATP5B protein is high and activated by reactive oxygen species (ROS) [36]. Overall, genes of the cyan module encode proteins that are activated by and protect from high levels of ROS.

Adult CF patients are sensitized to chronic opportunistic airway infections. In the blood transcriptome dataset, *P. aeruginosa* chronic infection negatively correlated with the magenta module. This correlation should be taken with caution because only two patients were not chronically infected by *P. aeruginosa*. Genes of the magenta module encode proteins important for erythrocyte differentiation and homeostasis, and for the hemoglobin complex. *SLC25A39*, a hub in this module, codes for a mitochondrial solute carrier protein. Silencing of the mouse Slc25a39 ortholog affected iron incorporation, essential for bacterial growth [37,38]. Thus, the magenta module points to the role of iron fixation in *P. aeruginosa* infections.

The present study has some limitations. We analyzed gene transcription in blood samples from 33 CF patients and 16 healthy controls. Confirmatory studies should be carried out in independent cohorts. We showed that blood is an informative surrogate tissue to address the contribution of inflammation to CFRD. However, evidence exists that this comorbidity also depends on an intrinsic pancreatic islet dysfunction whose study requires access to pancreas samples. Finally, in the future, patients should be followed longitudinally to correlate the gene signatures with the progression of the disease.

## Conclusions

In summary, using blood samples from CF patients, we identified modules of co-expressed genes that belong to relevant biological pathways. Detailed inspection of three modules that correlated with the presence of diabetes and lung function pointed to cell adhesion, leukocyte trafficking and production of ROS as central mechanisms in CFRD and pulmonary function decline. A fourth module that correlated with *P. aeruginosa* infection comprised genes important for iron fixation. Of note, we showed that blood is an informative surrogate tissue to address the contribution of inflammation to lung disease and diabetes in CF patients. Finally, we provided evidence that WGCNA is much valuable to analyze–omic datasets in rare genetic diseases as patient cohorts are inevitably small.

## Supporting information

**S1 Fig. Biotype distribution and expression level of the corresponding transcripts.**
(TIFF)

**S2 Fig. Network representation of the magenta module.** Top genes and their connections were visualized with Cytoscape [16]. Genes of interest were emphasized. The correlated clinical trait and main gene ontology (GO) term are shown.
(TIFF)

**S1 Table. Primer sequences and conditions used for qPCR validation.**
(DOCX)

**S2 Table. Differentially expressed genes between CF patients and controls.**
(DOCX)

**S3 Table. GO terms for differentially expressed genes between CF patients and controls.** (DOCX)

**S4 Table. Genes and modules.** Rows correspond to genes. A total of 15077 genes with FPKM > 0.1 in blood samples are listed. Columns list the gene name followed by module membership (kMEi) and corresponding p-values for each module of co-expressed genes. Lists of genes with high module membership can be sorted by selecting decreasing kMEi or increasing p-values. The kMEi is the correlation between the expression of a gene and the module eigengene. It ranges between 0 and 1. A gene is highly connected to other genes of a module when its kMEi approaches 1.
(XLSX)

## Acknowledgments

We are greatly indebted to cystic fibrosis patients, healthy volunteers and to the medical and paramedical staff of Montpellier, Hyères, Nice, and Toulouse CF Centers (CRCM) for their contribution to the MethylCF cohort. We thank Philippe Clair (Montpellier University/Biocampus Montpellier GenomiX facilities) for technical assistance and Madeleine Scott (Hendrix College, AR, USA) for English editing.

## Author Contributions

**Conceptualization:** Albertina De Sario.

**Data curation:** Mireille Claustres, Albertina De Sario.

**Formal analysis:** Fanny Pineau, Milena Magalhães, Enora Fremy, Abdillah Mohamed, Albertina De Sario.

**Funding acquisition:** Albertina De Sario.

**Investigation:** Davide Caimmi, Laurent Mely, Sylvie Leroy, Marlène Murris, Raphael Chiron.

**Methodology:** Davide Caimmi.

**Project administration:** Mireille Claustres, Albertina De Sario.

**Resources:** Davide Caimmi, Laurent Mely, Sylvie Leroy, Marlène Murris, Raphael Chiron.

**Supervision:** Albertina De Sario.

**Validation:** Fanny Pineau, Milena Magalhães.

**Writing – original draft:** Fanny Pineau, Albertina De Sario.

**Writing – review & editing:** Albertina De Sario.

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
