## [Decision Letter · Decision Letter 0]

7 Feb 2020

PONE-D-20-00190

Blood co-expression modules identify potential modifier genes of diabetes and lung function in cystic fibrosis

PLOS ONE

Dear Dr De Sario,

Thank you for submitting your manuscript to PLOS ONE. After careful consideration, we feel that it has merit but does not fully meet PLOS ONE’s publication criteria as it currently stands. Therefore, we invite you to submit a revised version of the manuscript that addresses the points raised during the review process.

Please follow the suggestions of both reviewers to improve your manuscript.

We would appreciate receiving your revised manuscript by Mar 22 2020 11:59PM. To enhance the reproducibility of your results, we recommend that if applicable you deposit your laboratory protocols in protocols.io, where a protocol can be assigned its own identifier (DOI) such that it can be cited independently in the future. For instructions see: http://journals.plos.org/plosone/s/submission-guidelines#loc-laboratory-protocols

We look forward to receiving your revised manuscript.

Kind regards,

Barbara Bardoni

Academic Editor

PLOS ONE

Journal Requirements:

Reviewers' comments:

Reviewer's Responses to Questions

**Comments to the Author**

1. Is the manuscript technically sound, and do the data support the conclusions?

Reviewer #1: Yes

Reviewer #2: Yes

2. Has the statistical analysis been performed appropriately and rigorously? 

Reviewer #1: Yes

Reviewer #2: Yes

3. Have the authors made all data underlying the findings in their manuscript fully available?

Reviewer #1: Yes

Reviewer #2: Yes

4. Is the manuscript presented in an intelligible fashion and written in standard English?

Reviewer #1: Yes

Reviewer #2: Yes

5. Review Comments to the Author

Reviewer #1: “Blood co-expression 1 modules identify potential modifier genes 2 of diabetes and lung function in cystic fibrosis”, by F. Pineau et al.

This study used transcriptome analysis of blood samples from individuals with CF in an attempt to find novel gene modifiers of CF disease. Using a Weighted Gene Correlation Network Analysis (WGCNA) approach, Lung function, BMI, diabetes presence and chronic P. aeruginosa infection were found to be correlated with modules of co-expressed genes. The study is clear and well presented. I have only a few minor comments which, if addressed, would render the article suitable for publication.

Comments:

1. Methods p.5-6 (Table 1): It would be informative to have the standard error (+/- SEM) for values in this table, rather than just the mean or median values, in order to better assess the similarity between the two patient sets.

2. Methods p.7 line 131: primer efficiency was determined to be 93% - presumably this means at least 93%?

3. Methods p. 7-8 (WGCNA section): this section is not very informative for readers who have not used this method of data analysis, and would benefit from a slight expansion to clarify some of the terms.

4. Results p 9-10: When presenting results for GO term enrichment a few of the most enriched groups are mentioned here along with a few of the representative genes. It would be more informative if the p values and number of genes in each group were shown in this section to reduce the necessity for looking at the supplementary table.

5. Results p 10 (Validation by real time PCR): the validation is well described and the results seem to be sound but the number of DE genes/lincRNAs validated is rather low, given the relative ease and rapidity of this technique. Particularly, given the non-validation of the lincRNA in the replication, I wonder why the authors did not try validating one of the other transcripts.

6. Results (WGCNA section including tables 3-4 and Figs 2-3): this section necessarily contracts the data in order to focus on a few modules of co-expressed genes that correlate with some CF parameters. However, it would be useful to provide a further clarification about the identity of other undiscussed modules which can be seen in Fig. 2. Table S4 is uninformative in this respect, as it is not easy to find the relevant genes in each module, even using the table legend on page 26. Either explain better how the data can be treated to find these genes, or provide another sheet in table S4 with the columns filtered to identify the important genes of each module.

7. Discussion p 16-19. The discussion of the suitability of the blood transcriptome for studying the contribution of DE gene expression in CFRD should be extended. In particular it would be interesting to know what proportions of which cell types were present in the whole blood from which the RNA samples were extracted. Some cytological data from similar blood samples to those used in the study would help in this respect, and would allow a fuller appreciation of the functional significance of the data.

Reviewer #2: In this manuscript, the authors performed RNA-seq analysis of blood samples obtained from patients affected with cystic fibrosis and healthy controls. They analyzed the data obtained by Weighted Gene Correlation Network Analysis and they found a correlation between lung function, body mass index, the presence of diabetes, and chronic P. aeruginosa infections with four modules of co-expressed genes.

This study is original and overall well written, I found the discussion a little bit lengthy. Figure 3 is too dense.

Figure legends are not included in the PDF file.

6. PLOS authors have the option to publish the peer review history of their article (what does this mean?). If published, this will include your full peer review and any attached files.

Reviewer #1: No

Reviewer #2: Yes: Enzo Lalli

---

## [Author Response · Author response to Decision Letter 0]

16 Mar 2020

Response to Reviewer #1:

This study used transcriptome analysis of blood samples from individuals with CF in an attempt to find novel gene modifiers of CF disease. Using a Weighted Gene Correlation Network Analysis (WGCNA) approach, Lung function, BMI, diabetes presence and chronic P. aeruginosa infection were found to be correlated with modules of co-expressed genes. The study is clear and well presented. I have only a few minor comments which, if addressed, would render the article suitable for publication.

Comments:

1. Methods p.5-6 (Table 1): It would be informative to have the standard error (+/- SEM) for values in this table, rather than just the mean or median values, in order to better assess the similarity between the two patient sets.

In Table 1 of the revised manuscript, we added the interquartile ranges (IQR).

2. Methods p.7 line 131: primer efficiency was determined to be 93% - presumably this means at least 93%?

Yes, it does mean at least 93%. Line 123 we added “at least”.

3. Methods p. 7-8 (WGCNA section): this section is not very informative for readers who have not used this method of data analysis, and would benefit from a slight expansion to clarify some of the terms.

Lines 135-152, we have summed up the main steps of WGCNA and slightly expanded the paragraph. 

4. Results p 9-10: When presenting results for GO term enrichment a few of the most enriched groups are mentioned here along with a few of the representative genes. It would be more informative if the p values and number of genes in each group were shown in this section to reduce the necessity for looking at the supplementary table.

Lines 193-197, we added the p-values and the number of genes for GO terms.

5. Results p 10 (Validation by real time PCR): the validation is well described and the results seem to be sound but the number of DE genes/lincRNAs validated is rather low, given the relative ease and rapidity of this technique. Particularly, given the non-validation of the lincRNA in the replication, I wonder why the authors did not try validating one of the other transcripts.

For real-time PCR validation, we selected 4 genes of interest having high, medium and low levels of expression. The lncRNA had rather low levels of expression in blood and was technically validated by real-time PCR in the discovery set of patients, the same used for RNAseq. In the confirmatory set of patients, although it did not reach significance, the lncRNA had the same direction of differential expression (overexpression in CF patients) and a similar fold-change as in RNAseq. 

6. Results (WGCNA section including tables 3-4 and Figs 2-3): this section necessarily contracts the data in order to focus on a few modules of co-expressed genes that correlate with some CF parameters. However, it would be useful to provide a further clarification about the identity of other undiscussed modules which can be seen in Fig. 2. Table S4 is uninformative in this respect, as it is not easy to find the relevant genes in each module, even using the table legend on page 26. Either explain better how the data can be treated to find these genes, or provide another sheet in table S4 with the columns filtered to identify the important genes of each module.

In the revised manuscript, we have added the KEGG analysis of the 28 modules and provided evidence that gene modules are biologically meaningful. See lines 222-225 and Table 3 showing the KEGG analysis of 28 modules.

In the revised legend of Table S4, we have specified how to identify the most connected genes of a module and we have better explained the kMEi (lines 554-560). We wish to keep Table S4 as a whole, including modules that were not analyzed in details and genes that are below our thresholds. This information can be useful for other studies or people interested in other genes.

7. Discussion p 16-19. The discussion of the suitability of the blood transcriptome for studying the contribution of DE gene expression in CFRD should be extended. In particular it would be interesting to know what proportions of which cell types were present in the whole blood from which the RNA samples were extracted. Some cytological data from similar blood samples to those used in the study would help in this respect, and would allow a fuller appreciation of the functional significance of the data.

In the result section, we have provided blood cell composition for a subset of CF patients. Blood cell composition was within normal ranges, as expected for CF patients without exacerbation. In addition, we have specified that whole blood samples had been gathered in PAXgene tubes that not only stabilize RNA, but also preserve all types of circulating leukocytes. This is not the case for other blood storage methods that alter the cell proportion in blood samples and also the transcriptomic profile. A bibliographic reference has been provided. See lines 162-169 and reference 18.

We preferred to treat the issue in the result section to avoid expanding the discussion in line with referee 2 comments.

Reviewer #2: In this manuscript, the authors performed RNA-seq analysis of blood samples obtained from patients affected with cystic fibrosis and healthy controls. They analyzed the data obtained by Weighted Gene Correlation Network Analysis and they found a correlation between lung function, body mass index, the presence of diabetes, and chronic P. aeruginosa infections with four modules of co-expressed genes.

This study is original and overall well written, I found the discussion a little bit lengthy. Figure 3 is too dense.

We thank the reviewer for his comments. The discussion was slightly shortened and a few paragraphs were rephrased. Figure 3 was split into figure 3 and figure 4.

Figure legends are not included in the PDF file.

Figure legends are embedded in the manuscript according to PLOS One guidelines.

6. PLOS authors have the option to publish the peer review history of their article (what does this mean?). If published, this will include your full peer review and any attached files.

Do you want your identity to be public for this peer review? For information about this choice, including consent withdrawal, please see our Privacy Policy.

Reviewer #1: No

Reviewer #2: Yes: Enzo Lalli

Figures 1 to 4 have been uploaded to the PACE tool.

---

## [Editor Report · Decision Letter 1]

20 Mar 2020

Blood co-expression modules identify potential modifier genes of diabetes and lung function in cystic fibrosis

PONE-D-20-00190R1

Dear Dr. De Sario,

We are pleased to inform you that your manuscript has been judged scientifically suitable for publication and will be formally accepted for publication once it complies with all outstanding technical requirements.

With kind regards,

Barbara Bardoni

Academic Editor

PLOS ONE
---

## [Editor Report · Acceptance letter]

7 Apr 2020

PONE-D-20-00190R1 

Blood co-expression modules identify potential modifier genes of diabetes and lung function in cystic fibrosis 

Dear Dr. De Sario:

I am pleased to inform you that your manuscript has been deemed suitable for publication in PLOS ONE. Congratulations! Your manuscript is now with our production department. 

With kind regards,

on behalf of

Dr. Barbara Bardoni 

Academic Editor

PLOS ONE